# Programme death ligand 1 expressions as a surrogate for determining immunotherapy in cervical carcinoma patients

**Sebastian A. Omenai**[¤]*, **Mustapha A. Ajani, Clement A. Okolo**

Department of Pathology, University College Hospital, Ibadan, Oyo State, Nigeria

¤ Current address: Department of Anatomical Pathology, Edo State University, Uzairue, Edo State, Nigeria
* sebeanom@gmail.com

## Abstract

### Background

The programme death ligand1 and its receptor (PD-1/PD-L1) interaction is a target for blockage by immunotherapy that uses the body's own immune system. Some studies show that PD-L1 expressing tumours are also more aggressive with poor prognosis. This study evaluated the immunohistochemical expression of PD-L1 in uterine cervical carcinomas. Women with cervical cancer would benefit from its use as a marker in therapy and prognosis.

### Methods

Hospital-based cross-sectional retrospective study was conducted. The study materials included 183 archived formalin fixed and paraffin embedded (FFPE) tissue blocks with histological diagnosis of cervical carcinoma diagnosed in our facility within a five-year period (January 2012 and December 2016) that met the study criteria. Data were extracted from records in the Department and immunohistochemistry was done using polyclonal antibodies to PD-L1 (GTX104763, Genetex). Obtained data were analysed using SPSS version 23. $P < 0.05$ was considered significant.

### Results

A hundred and eighty-three cases of cervical cancer were studied. PD-L1 was positive in 57.4% of all cases. The diffuse pattern of staining was the major pattern accounting for 88.5% of positive cases. Poorly differentiated cervical carcinomas are less likely to express PD-L1. Within the histologic types, the squamous cell carcinomas expressed PD-L1 in 58.7%, and 50% of adenocarcinomas were positive. PD-L1 was not expressed in all cases of adenoid cystic carcinomas and basaloid squamous cell carcinomas.

### Conclusion

A significant population of cervical carcinoma expresses PD-L1 by immunohistochemistry. PD-L1 prevalence is lower amongst the poorly differentiated cancers compared to other grades.

**Data Availability Statement:** All relevant data are within the manuscript.

**Funding:** The authors received no specific funding for this work.

**Competing interests:** The authors have declared that no competing interests exist.

## Introduction

Uterine cervical carcinoma is the fourth most common cancer amongst women worldwide and the most common gynaecological cancer in sub-Saharan African [1–3]. In Nigeria, the incidence of cervical cancer is still significantly high, second only to breast cancer as the leading cause of cancer death [2]. Cervical cancer is caused by persistent infection with high risk Human Papilloma Virus (HPV) with types 16 and 18 accounting for more than 70% of invasive cervical carcinomas and types 16, 18, 35 and 45 being the commonest amongst sub-Saharan Africans [4–6]. Cervical squamous cell carcinomas are graded using the Broder's classification into well differentiated (grade 1), moderately differentiated (grade 2) and poorly differentiated (grade 3), while cervical adenocarcinomas are graded based on architectural features [7]. This stratification of cervical carcinomas have not provided consistent prognosis or predicted response to therapy [8]. Currently management of cervical cancers includes the use of immunotherapies following the approval of anti-PD1 (Pembrolizumab). The use of cancer immunotherapy is due to better understanding of the tumour microenvironment. Inhibition of PD-1/PD-L1 is one way to restore host immunity against cancer cells with the prospects of durable remissions [9,10]. The body has a natural protective mechanism against cancer by recognizing such malignant cells as foreign and attacking and destroying them a process which can auto-propagate as the cancer immunity cycle. Cancer arises when it successfully evades the immune system by ensuring that that the immune system fails to recognize the malignant cells as foreign [11]. Cancer cells expressing PD-L1 binds to the PD-1 receptor on the surface of cytotoxic T-cells and inactivates the T-cells thus preventing immune response against the cancer cells [9,11].

Immunohistochemical expression of PD-L1 has been associated with more aggressive cervical carcinomas with poorer prognosis despite its predictive value to the response of immunotherapy [12–14]. Loharamtaweethong et al. in their study showed that PD-L1 immunohistochemical expression was not associated with worse outcomes in cervical carcinomas [15]. It has been demonstrated in a study that squamous cell carcinoma with marginal pattern of PD-L1 staining had unambiguous survival benefits compared to the diffusely staining pattern [16]. This could actually be due to the fact that these type of carcinomas have high density of CD8+ tumour infiltrating lymphocytes (TILs) [17,18]. The presence of prominent TILs in PD-L1 positive cervical carcinomas is associated with longer survival compared to cervical carcinoma patients with low or no TILs [19,20]. Thus, care is needed when interpreting PD-L1 positivity in relation to prognosis, this has led to development and proposition of the immune-score in predicting clinical outcomes in cervical carcinoma [21]. Howitt et al. demonstrated that chromosome 9p24.1 gene copy number alterations are responsible for increased PD-L1 expression in a subset of cervical carcinomas [22]. Rotman et al. in their study suggested that the heterogenous nature of PD-L1 expression by immunohistochemistry makes it unsuitable for patient selection [23].

This study aims to provide data on the pattern of immunohistochemical expression of PD-L1 in cervical carcinomas as seen in the University College Hospital, Ibadan, with a view to determining usefulness of PD-L1 immunostaining patterns in determining selection for immunotherapy.

## Methods

### Ethical consideration

Written ethical clearance was obtained from the Joint Ethical Review Committee of the College of Medicine, University of Ibadan and the University College Hospital, Ibadan (UI/EC/

17/0137). All cases were treated as anonymous. Data collected were stored in a laptop that was secured by a password.

## Study design

This study is a retrospective cross-sectional study involving the review of all the histologically diagnosed uterine cervical carcinomas in the Department of Pathology, University College Hospital, Ibadan from January 2012 to December 2016. The histologically diagnosed carcinomas and the relevant clinical and histopathological information relating to these tumours were extracted from the surgical daybook records of the Department. All cases of cervical carcinoma diagnosed during the study period irrespective of histological type were included in the study. Cases whose paraffin blocks were missing or with insufficient diagnostic tissue were excluded from the study.

Immunohistochemistry was done following the manufacturer's protocol. Polyclonal antibody for PD-L1 is anti PD-L1 GTX104763 rabbit antibody (Genetex, Taiwan) used at 1:500 dilutions. The slides were incubated with the primary antibody, rabbit anti-PD-L1 antibody in a humidified chamber at room temperature for one hour, and then incubated for 15 minutes with MACH 4$^{TM}$ mouse probe (Biocare medicals) at room temperature. MACH 4$^{TM}$ HRP-polymer [horseradish peroxidase polymer] (Biocare medicals) was added to the slides and allowed to incubate for 15 minutes at room temperature, and then washed using wash buffer. The DAB [3,3 diaminobenzidine] (Biocare medicals) chromogen substrate was added next and allowed to incubate for seven minutes. The slides were counterstained with Haematoxylin for ten seconds at room temperature. The antibody is visualized as membrane staining. The combined positive score (CPS) was used in assessing the PD-L1 expression. CPS counts the number of positive malignant cells (show complete and/or partial circumferential linear plasma membrane staining), lymphocytes and macrophages divided by the overall viable tumour cell number multiplied by 100 [24]. Cervical cancers were regarded as expressing PD-L1 positivity if CPS is ≥1%. Negative PD-L1 expression is regarded as <1%. A distinction was made between tumours that stained more at the margins (tumour nest-stroma interface) and those that had diffuse staining of tumour nests.

## Data analysis

The data obtained were entered into an excel spreadsheet (Microsoft, Redmond, Washington, United State). The analysis was done using IBM SPSS statistics (version 23 IBM Corporation, Armonk, New York). The level of significance was set as p < 0.05. Categorical data (histological diagnosis, PD-L1 immunohistochemical expression) were recorded as frequencies. Kruskal Wallis H test was used to test for a relationship between PD-L1 status and the histological grade of cervical cancers.

## Results

Out of 276 cases of cervical carcinomas within the study period only 183 cases had enough diagnostic tissue in the FFPE blocks for immunohistochemical analysis and were used for the study. Out of the 183 cases of cervical carcinomas studied, 155 (84.7%) were squamous cell carcinomas, 12 cases (6.6%) were adenocarcinomas, 11 cases (6.0%) were adenosquamous carcinomas and the remaining 5 cases (2.7%) had adenoid cystic carcinomas. Both squamous cell carcinomas and adenocarcinomas peaked in the 6$^{th}$ decade (Figs 1 and 2).

From the 183 cases studied 105 (57.4%) were positive for PD-L1 immunohistochemical stain and majority (88.5%) of these positive cases showed diffuse pattern of staining nest of malignant cells (Fig 3) while 11.5% of the positive cases showed mainly marginal pattern of

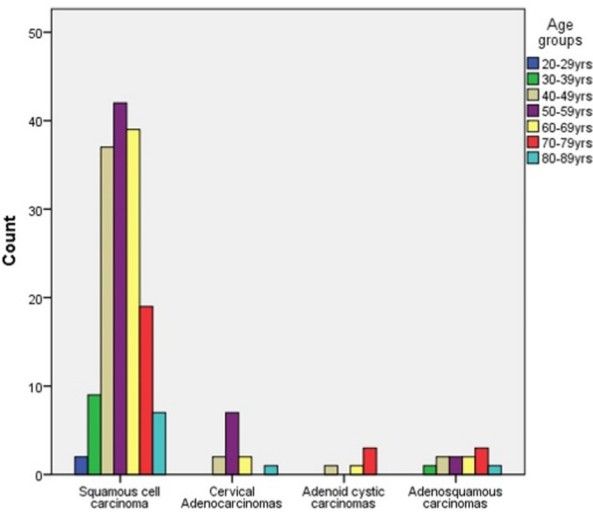

**Fig 1. Clustered bar chart showing age group and the histological types of cervical carcinoma.**

staining, [where the malignant cells at the nest–stromal interface were majorly the positive cells] (Fig 4). Amongst the squamous cell carcinomas, 58.7% were positive for PD-L1, while 50.0% of all the adenocarcinomas and 72.7% of all the adenosquamous carcinomas were

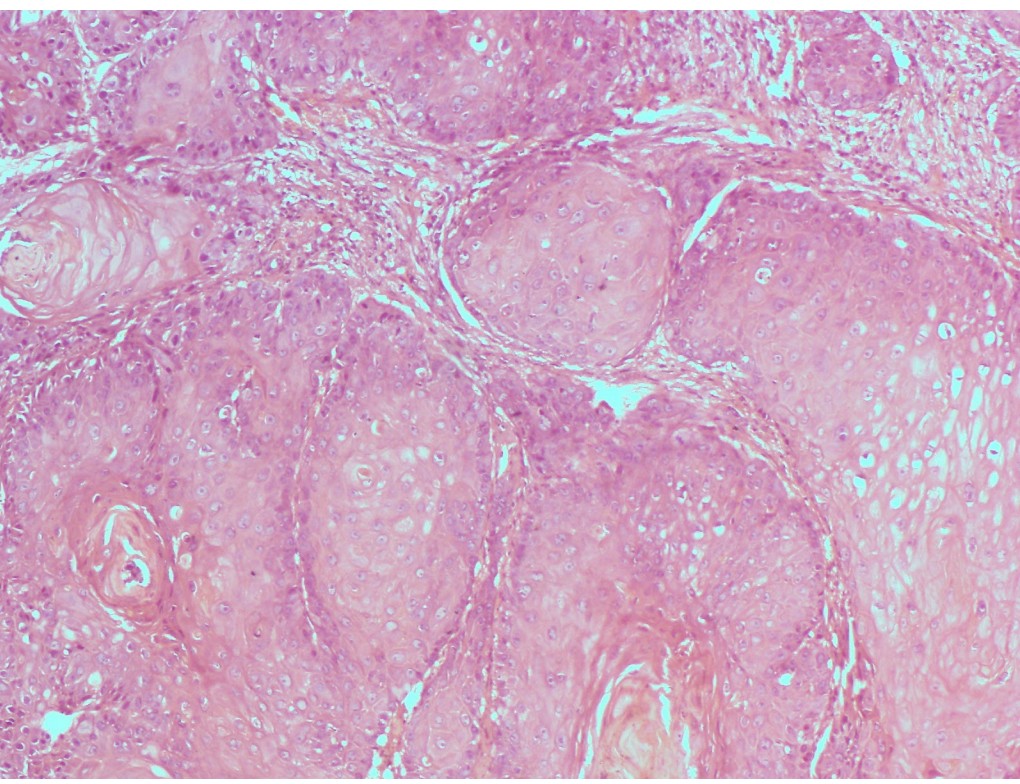

**Fig 2. Photomicrograph showing well differentiated large cell keratinizing squamous cell carcinoma (Haematoxylin and eosin stains, X100).**

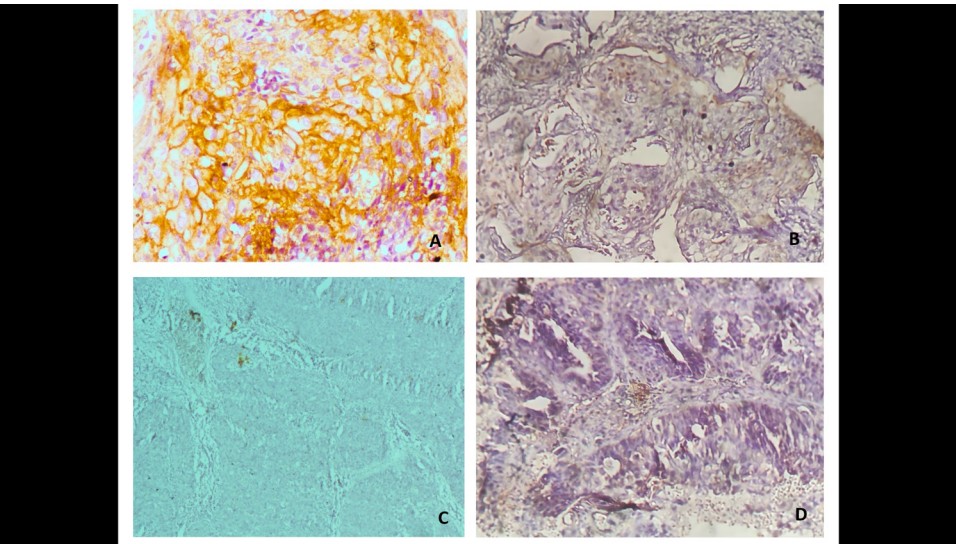

**Fig 3. (a)** Photomicrograph showing diffuse staining in a large cell keratinizing squamous cell carcinoma (anti-PD-L1 immunohistochemical stain, X400) **(b)** weak and partial membranous staining (anti-PD-L1 immunohistochemical stain, X200) **(c)** Negative PD-L1 staining **(d)** PD-L1 positive adenocarcinoma (anti-PD-L1 immunohistochemical stain, X200).

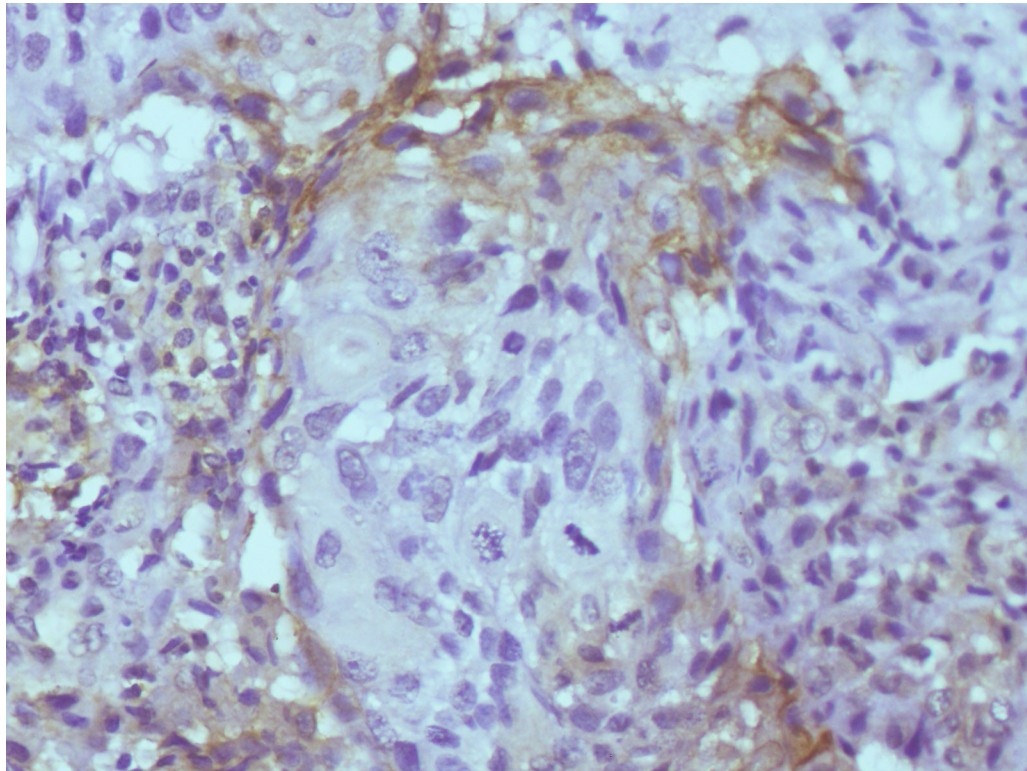

**Fig 4. Photomicrograph showing marginal staining of nest of invasive squamous cell carcinoma (anti-PD-L1 Immunohistochemical stain, X100).**

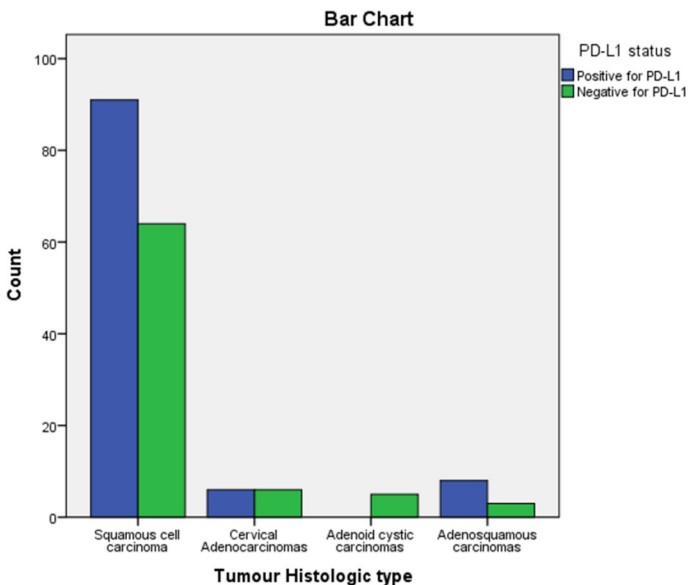

**Fig 5. Bar chart showing the correlation between the PD-L1 status and histological types of cervical carcinoma.**

positive with PD-L1 immunohistochemical stain. The five cases of adenoid cystic carcinomas were all negative for PD-L1 immunohistochemical stain. There was no significant statistical difference between expression of PD-L1 and histological type of cervical carcinomas (Fig 5).

Diffuse staining of malignant cells was seen in 86.7% (78) of squamous cell carcinomas while 13.3% (12) had marginal staining pattern. The majority (91.7%) of these squamous cell carcinomas demonstrating marginal stains were large cell non-keratinizing variant. Only one case from the nine PD-L1 positive large cell keratinizing squamous cell carcinoma showed the marginal pattern of staining. Adenocarcinomas and adenosquamous carcinomas showed only diffuse staining pattern.

Within the variants of squamous cell carcinomas, PD-L1 was positive in 60.0% of the large cell keratinizing cases, while the large cell non-keratinizing variant had 59.9% of them expressing PD-L1. None of the basaloid squamous cell carcinoma expressed PD-L1. The mucinous and clear cell variants of cervical adenocarcinoma did not express PD-L1.

Poorly differentiated carcinomas were only positive for PD-L1 in 42.3% of cases, while moderately differentiated carcinomas were positive for PD-L1 in 65.3% of the cases and 57.6% of well differentiated carcinomas were positive for PD-L1. A Kruskal-Wallis H test showed that there was a statistically significant difference in PD-L1 status between the different drug histological grades, $\chi 2$ (2) = 7.308, p = 0.026, with a mean rank PD-L1 status of 91.82 for well differentiated, 84.74 for moderately differentiated and 105.79 for poorly differentiated carcinomas (Table 1).

**Table 1. Relationship between PD-L1 expression and histological grades of cervical carcinomas.**

| | Variable | Histological Grade (%) | | | $X^2$ | p-value |
|---|---|---|---|---|---|---|
| | | Well differentiated carcinomas | Moderately differentiated carcinomas | Poorly differentiated carcinomas | | |
| 1 | **PD-L1 Status** | | | | | |
| | Positive for PD-L1 | 19 (57.6%) | 64 (65.3%) | 22 (42.3%) | 7.308 | **0.026** |
| | Negative for PD-L1 | 14 (42.4%) | 34 (34.7%) | 30 (57.7%) | | |
| 2 | **Pattern of PD-L1 expression** | | | | | |
| | Diffuse staining of tumour cells | 17 (89.5%) | 55 (87.3%) | 20 (90.9%) | 0.231 | 0.891 |
| | Marginal staining of tumour nests | 2 (10.5%) | 8 (12.7%) | 2 (9.1%) | | |

## Discussion

Majority of cases of cervical carcinomas are squamous cell carcinoma and adenocarcinoma. Significant proportion of these two histological types are more likely to express PD-L1 by immunohistochemistry compared to the rare histological types of cervical cancer. The poorly differentiated grades of cancers are less likely to be positive for PD-L1 immunohistochemical expression.

This study showed that PD-L1 was positive in 57.4% of all uterine cervical carcinomas which is relatively similar to the finding of 51.6% in a study by Zhang et al., and 43.9% by Heeren et al. [16,25]. Enwere et al. reported 71% positive PD-L1 in their study which is significantly higher than our study and closer to those of Loharamtaweethong et al. and Chung et al. that reported 83% and 83.7% respectively [15,26,27]. A significant proportion of cervical cancers in our environment express PD-L1 and are likely to be aggressive with poor prognosis and would potentially benefit from anti-PD-1 immunotherapy. Although the use of PD-L1 for prognosis in cervical cancer is currently disputable as some studies reported positive correlation of poor outcome with PD-L1 expression [13,14]. Other studies such as the one by Loharamtaweethong could not establish worse outcomes with PD-L1 expression [15]. Another plausible explanation for this discordance in using PD-L1 expression as a prognostic marker can be the finding by Feng et al. that the presence of prominent tumour infiltrating lymphocytes was associated with clear trends towards longer survival [19].

Amongst the histological types, 58.7% of all squamous cell carcinomas were positive for PD-L1 which is comparable to findings by Reddy et al., Heeren et al. and Saglam et al. that reported findings of 37.8%, 54% and 35% respectively [16,17,28]. PD-L1 was positive in 50.0% of all cases of adenocarcinomas, which is significantly higher compared to findings of Reddy et al. which was 7%, Heeren et al. which was 14% and Saglam et al. which was 17% [16,17,28]. The differences might be due to heterogeneity of PD-L1 expression [29], and the fact that most of these other studies were done using microarray techniques (PD-L1 has shown heterogenous expression in cervical cancer), and tumour proportion score as against combined positive score. Also, the relatively fewer number of cervical adenocarcinomas seen in this study might also be responsible for the relatively higher PD-L1 expression. PD-L1 was positive in 72.7% of adenosquamous carcinoma which is higher when compared to the findings by Reddy et al. that reported a frequency of 28.6% [28]. This difference might be due to the few cases of adenosquamous carcinomas seen in this study as well the fact that Reddy et al used tumour proportion score and a higher percentage in accessing PD-L1 status. The trend of having more cases of PD-L1 expression in adenosquamous carcinomas compared to pure adenocarcinomas were similar in both studies.

PD-L1 was expressed more in lower grade carcinomas with 57.6% and 65.3% of well differentiated and moderately differentiated tumours respectively, with only 42.3% of poorly differentiated carcinomas. Similar result was reported by Reddy et al., they observed that only 6.7% of grade 3 squamous cell carcinoma expressed PD-L1 [28]. These observations in cervical carcinomas contradict the findings of Lin et al. that reported high grade tumours to have more PD-L1 positive expression in oral squamous cell carcinomas [30]. It is also documented that PD-L1 is one of the mechanism of immune escape by cervical carcinomas and that such tumours are associated with poor prognosis [31,32]. One would have expected that higher grade carcinomas will express a marker of poor prognosis more than the well differentiated carcinomas. So, the finding of fewer PD-L1 positive cases among the poorly differentiated carcinomas in this present study. This could be part of the reason why the use of PD-L1 as a prognostic marker has failed to provide a consistent and reproducible result. It could also be due to

similar reasons why the histological grades of cervical carcinomas have not provided consistent prognostic value [7,8].

Diffuse staining pattern (88.5%) was the most common immunohistochemical staining pattern in this present study which was similar to the findings of Heeren et al. [16]. Marginal staining pattern accounted for 11.5% and was seen only in squamous cell carcinomas and 91.7% of the cases that showed marginal pattern of staining are the large cell non-keratinizing variant. Although this was not a statistically significant finding, the marginal staining pattern and large cell non keratinizing variant has been associated with better prognosis [16,17,32–34].

The use of PD-L1 as biomarker is not without other challenges as there are currently many clones of this antibody with many of them being research only, laboratory non-diagnostic clones. Some of the common clones E1L3N, E1J2J, SP142, 28–8, 22C3, SP263 and 5H1. Genetex manufactures the clones GTX104763, HL1041which was used in this study, for non-clinical purposes and it is one of the least popular clones. It has shown good results as seen in other published studies [30,35]. Clone 22C3 pharmDX assay is approved as a companion diagnostic for Pembrolizumab in the management of advanced or recurrent cervical cancer by regulatory authorities using combined positive score (CPS) of 1 or greater.

## Limitations

This study has a number of limitations, firstly the usage of a non-clinical for laboratory only use clone is a major limitation as this is not the approved clone for cervical patient care. We used this clone as it was readily accessible to our laboratory. Despite this limitation, we believe that our observations using the Genetex clone provide an insight into the applicability of PD-L1 by immunohistochemistry in cervical cancer patients in this environment.

Secondly the number of cases were relatively few especially with some variants of squamous cell carcinoma. This reduces the power of inference of our observations. Thirdly, record keeping with incomplete data and data loss contributed to the significantly reduced number of cases meeting the inclusion criteria. A prospective study would have been more ideal and this could have as well eliminated the factor of the storage condition of formalin fixed and paraffin embedded archival tissue blocks. Despite these limitations the outcome of this study provide data on the immunohistochemical expression of PD-L1 on variants of cervical carcinomas.

## Conclusion

PD-L1 expression is seen in a significant number of cervical carcinomas with a significantly lesser expression in poorly differentiated carcinomas. The basaloid variant of squamous cell carcinoma and adenoid cystic carcinoma did not express PD-L1 by immunohistochemistry. The most common pattern of staining is the diffuse pattern. The marginal staining pattern was seen only in the squamous cell carcinomas. Nigerian women with cervical cancers would likely benefit tremendously with immunotherapy treatment as this study has shown that PD-L1 pathway play a role in cervical cancer development in a significant proportion of the study population.

## Author Contributions

**Conceptualization:** Sebastian A. Omenai, Clement A. Okolo.

**Data curation:** Sebastian A. Omenai.

**Formal analysis:** Sebastian A. Omenai.

**Investigation:** Sebastian A. Omenai.

**Methodology:** Sebastian A. Omenai.

**Project administration:** Mustapha A. Ajani.

**Supervision:** Mustapha A. Ajani, Clement A. Okolo.

**Validation:** Sebastian A. Omenai, Clement A. Okolo.

**Writing – original draft:** Sebastian A. Omenai.

**Writing – review & editing:** Mustapha A. Ajani, Clement A. Okolo.

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
