## [Decision Letter · Decision Letter 0]

10 Dec 2021

PONE-D-21-26179Programme Death Ligand 1 Expressions as a Surrogate for Determining Immunotherapy in Cervical carcinoma PatientsPLOS ONE

Dear Dr. Omenai,

Thank you for submitting your manuscript to PLOS ONE. After careful consideration, we feel that it has merit but does not fully meet PLOS ONE’s publication criteria as it currently stands. Therefore, we invite you to submit a revised version of the manuscript that addresses the points raised during the review process. Please note that both reviewers have raised some concerns regarding the data analysis. Kindly revise the manuscript addressing these issues.

Please submit your revised manuscript by 15 Januaru 2022.  If you will need more time than this to complete your revisions, please reply to this message or contact the journal office at plosone@plos.org. Please include the following items when submitting your revised manuscript:A rebuttal letter that responds to each point raised by the academic editor and reviewer(s). You should upload this letter as a separate file labeled 'Response to Reviewers'.A marked-up copy of your manuscript that highlights changes made to the original version. You should upload this as a separate file labeled 'Revised Manuscript with Track Changes'.An unmarked version of your revised paper without tracked changes. You should upload this as a separate file labeled 'Manuscript'.

We look forward to receiving your revised manuscript.

Kind regards,

Afsheen Raza, PhD

Academic Editor

PLOS ONE

Journal Requirements:

2. In your ethics statement in the manuscript and in the online submission form, please ensure that you have discussed whether all data/samples were fully anonymized before you accessed them and/or whether the IRB or ethics committee waived the requirement for informed consent. If patients provided informed written consent to have data/samples from their medical records used in research, please include this information.

Reviewers' comments:

Reviewer's Responses to Questions

**Comments to the Author**

1. Is the manuscript technically sound, and do the data support the conclusions?

Reviewer #1: Yes

Reviewer #2: Partly

2. Has the statistical analysis been performed appropriately and rigorously? 

Reviewer #1: Yes

Reviewer #2: Yes

3. Have the authors made all data underlying the findings in their manuscript fully available?

Reviewer #1: Yes

Reviewer #2: Yes

4. Is the manuscript presented in an intelligible fashion and written in standard English?

Reviewer #1: Yes

Reviewer #2: Yes

5. Review Comments to the Author

Reviewer #1: This is an interesting paper and a lot of work has been done. However, there are many points to be clarified and improved upon:

1. The use of the English language and grammar should be improved.

2. Mention in the abstract the number of cases stained and scored. Mention the way the scoring was performed. mention links with clinical parameters. Ad which PDL1 clone was used.

3. Give a clear representation of the literature on PDL1 expression in cervical cancer. There are papers missing. Moreover, there is conflicting data on relation with prognosis and relation with amplification of the PDL1 locus. Please, elaborate and give cite all literature on cervical cancer.

4. PD-L1 is only one of the immune evasion mechanisms. The way the abstract is written at the moment it seems like it is the only one. Please, get more nuance into the discussion of the results.

5. Do comment on the various clones targeting PDL1 used for IHC. Is the clone the authors used also used by others or how does it relate to the standard clones used. Please, give more information. It is a big problem and should be addressed in the discussion. Please, show representative cases of positive, weak and negative expression in the SCC and AC histologies.

6. Was there other clinical data available on this patient cohort? HPV type, metastatic status? It would be interesting to compare expression percentages and expression patterns to other clinical data and survival.

7. Can the authors say something about tumor infiltrating myeloid cells being positive for PDL1? Or were there cases with high PDL1 expression on cells in the stromal compartment?

8. It is known that patients with expression PDL1 can also be unresponsive to PD1/PDL1 checkpoint inhibition. Please, tone down the discussion and also add literature on PD1/PDL1 being not effective and PD1/PDL1 being a biomarker with limited value.

Reviewer #2: Thank you for submitting this very interesting manuscript. The manuscript is well written and highlights the importance of PD-L1 testing in advanced cervical cancers, in particular the different rates of positivity based on histopathological characteristics. I have one major comment - and this is with regards to measuring the PD-L1 score. TPS is currently only used for non-small cell lung cancers, whereas other sites utilise the CPS scoring system. Your research has calculated the TPS score for PD-L1, which is a major flaw, in my opinion; and I wonder if the results would differ if the CPS scoring system was applied instead. The paper will need to be rewritten after recalculating the PD-L1 scores based on the CPS scoring system. In addition, please indicate if PD-L1 22C3 testing was used or SP142 assay.

6. PLOS authors have the option to publish the peer review history of their article (what does this mean?). If published, this will include your full peer review and any attached files.

Reviewer #1: No

Reviewer #2: No

---

## [Author Response · Author response to Decision Letter 0]

13 Jan 2022

Reviewer #1.

We have made some spelling and grammar corrections in the manuscript just like suggested. The number of cases stained and scored as well as the PD-L1 clone used are now mentioned in the abstract. We have added more literature on the use of PD-L1 in cervical carcinomas especially in regards to prognostic implications of PD-L1 expression. Truly, PD-L1 is just one of the immune evasion mechanisms of cancers currently being studied. Our write up initially inadvertently presented PD-L1 as the only mechanism, which we have corrected in the manuscript. We agree that using non diagnostic PD-L1 clone is a major drawback, we had difficulty sourcing for the more popular clones for our laboratory. We have expressed the limitations this pose to our findings in the manuscript. We reviewed the slides and used the combine positive scoring system which takes note of positive infiltrating myeloid cells in determining PD-L1 status.

Reviewer #2.

We have recalculated the PD-L1 status using the CPS. We didn’t use 22C3 or SP142 clone assay in our study. Hopefully we can get dealers to supply to our laboratory as soon as we establish the need for PD-L1 immunohistochemistry in our society. We have stated the non-diagnostic genetex PD-l1 assay we used and the subsequent limitations it poses.

---

## [Editor Report · Decision Letter 1]

24 Jan 2022

Programme Death Ligand 1 Expressions as a Surrogate for Determining Immunotherapy in Cervical carcinoma Patients

PONE-D-21-26179R1

Dear Dr. Omenai,

We’re pleased to inform you that your manuscript has been judged scientifically suitable for publication and will be formally accepted for publication once it meets all outstanding technical requirements.

Kind regards,

Afsheen Raza, PhD

Academic Editor

PLOS ONE
---

## [Editor Report · Acceptance letter]

26 Jan 2022

PONE-D-21-26179R1 

Programme Death Ligand 1 Expressions as a Surrogate for Determining Immunotherapy in Cervical carcinoma Patients 

Dear Dr. Omenai:

I'm pleased to inform you that your manuscript has been deemed suitable for publication in PLOS ONE. Congratulations! Your manuscript is now with our production department. 

Kind regards, 

on behalf of

Dr. Afsheen Raza 

Academic Editor

PLOS ONE